# Common Mode Component and Its Potential Effect on GPS-Inferred Three-Dimensional Crustal Deformations in the Eastern Tibetan Plateau

**Yuanjin Pan [1] , Ruizhi Chen [1,2,] *, Hao Ding [3] , Xinyu Xu [3] , Gang Zheng [4] , Wenbin Shen [3] , YiXin Xiao [3] and Shuya Li [3]**

[1]   State Key Laboratory of Information Engineering in Surveying, Mapping and Remote Sensing, Wuhan University, Wuhan 430079, China
[2]   Collaborative Innovation Center of Geospatial Technology (INNOGST), Wuhan 430079, China
[3]   School of Geodesy and Geomatics, Wuhan University, Wuhan 430079, China
[4]   GNSS Research Center, Wuhan University, Wuhan 430079, China
*   Correspondence: ruizhi.chen@whu.edu.cn

**Abstract:** Surface and deep potential geophysical signals respond to the spatial redistribution of global mass variations, which may be monitored by geodetic observations. In this study, we analyze dense Global Positioning System (GPS) time series in the Eastern Tibetan Plateau using principal component analysis (PCA) and wavelet time-frequency spectra. The oscillations of interannual and residual signals are clearly identified in the common mode component (CMC) decomposed from the dense GPS time series from 2000 to 2018. The newly developed spherical harmonic coefficients of the Gravity Recovery and Climate Experiment Release-06 (GRACE RL06) are adopted to estimate the seasonal and interannual patterns in this region, revealing hydrologic and atmospheric/nontidal ocean loads. We stack the averaged elastic GRACE-derived loading displacements to identify the potential physical significance of the CMC in the GPS time series. Interannual nonlinear signals with a period of ~3 to ~4 years in the CMC (the scaled principal components from PC1 to PC3) are found to be predominantly related to hydrologic loading displacements, which respond to signals (El Niño/La Niña) of global climate change. We find an obvious signal with a period of ~6 yr on the vertical component that could be caused by mantle-inner core gravity coupling. Moreover, we evaluate the CMC's effect on the GPS-derived velocities and confirm that removing the CMC can improve the recognition of nontectonic crustal deformation, especially on the vertical component. Furthermore, the effects of the CMC on the three-dimensional velocity and uncertainty are presented to reveal the significant crustal deformation and dynamic processes of the Eastern Tibetan Plateau.

**Keywords:** common mode component (CMC); GPS-inferred three-dimensional velocity; seasonal and interannual signals; geophysical interpretation; GRACE-modeled elastic deformation

## 1. Introduction

As the Earth's "Third Pole" with an average elevation of ~5000 m, the Tibetan Plateau is evidently important to the balance of global climate change. The uplift of the Tibetan Plateau and the migration of the materials therein are characterized by complicated crustal deformation and dynamic processes involving surface mass redistribution, deep lithosphere motion, and flow in the weakened lower crust [1,2]. The Eastern Tibetan Plateau (ETP), a complex geological tectonic setting and seismically active area, plays a crucial role in the movement and evolution of the Tibetan Plateau. Depicting the crustal deformation and uplift of the ETP is very important for developing a significant understanding of the India–Asia and continent–continent collisions [3,4]. However, the mechanisms responsible for

the tectonic uplift and dynamic processes of the ETP remain relatively unknown despite the numerous studies that have been presented on this topic [2,5–7].

In recent years, an increasing number of geodesists and geophysicists have studied the vertical crustal deformation of the Tibetan Plateau and its forelands [4,8–10]. One of the key issues is determining how to separate surface nontectonic deformation (e.g., surface hydrologic, atmospheric, and nontidal ocean loading displacements) from tectonic deformation to reveal the characteristics of vertical tectonic deformation. Surface elastic deformation and lithosphere motion respond to mass redistribution within the Earth's fluid envelope through deep tectonics and flows within the mantle [11]. Measurements from remote sensing, including those from the Global Positioning System (GPS) and Gravity Recovery and Climate Experiment (GRACE), are sensitive to ground motion and gravity anomalies, which contain complex geological signals that have been routinely used in geophysics. GPS-derived velocities have been widely used to analyze the crustal deformation of the Tibetan Plateau, revealing the dynamic mechanisms of the plate motion therein [6,12]. In addition, the vertical component of GPS time series was applied to a seasonal hydrology loading study [3,8]. Furthermore, GRACE-delivered gravity data have been widely used in Earth science since 2002 [13]. For example, water storage and ice mass changes in the Tibetan Plateau can be monitored by using GRACE-modeled data [14,15]. Understanding the processes that generate these geophysical signals in the Tibetan Plateau can be approached with a signal spectrum analysis of geodetic time series.

The fact that the common mode components (CMCs) in regionally dense GPS time series remain unmodeled and unresolved in geophysical research constitutes another popular issue. A CMC is considered to consist of a series of unmodeled geophysical processes, including environmental load effects and technically relevant systematic errors that remain after GPS processing [16]. Most previous studies focused on how to extract a CMC from GPS time series and the effect of CMCs on GPS-derived velocities [3,17]. However, few studies have analyzed the geophysical mechanism of CMCs in the time domain. For nonstationary signals, the average amplitude may be obtained with the Fourier spectrum. However, the time-varying characteristics of such signals, which may be more significant for the interpretation of geophysical phenomena, cannot be identified. In this study, we analyze nonlinear signals and three-dimensional crustal deformation throughout the ETP. CMCs are decomposed from 69 continuous GPS time series in the ETP according to the principal component analysis (PCA) method. The wavelet time-frequency spectrum is adopted to identify the interannual oscillation of nonlinear CMC signals. Furthermore, we present the GRACE-derived displacements induced by the surface hydrology, atmosphere and nontidal modeled surface loading to reveal significant oscillation signals throughout the ETP. Finally, we evaluate the influences of CMCs on 3-D velocity vectors in the ETP.

## 2. Materials and Methods

### 2.1. Continuous GPS Observations

In this study, we used a dense concentration of GPS continuous observation stations throughout the ETP; the GPS network comprises 6 long-term observation stations covering the period from 1999 to 2018 and 63 short-term observation stations covering the period from 2010 to 2018, as shown in Figure 1. All raw GPS observation data were obtained from the Crustal Movement Observation Network of China (CMONOC-I and CMONOC-II). The daily station positions were processed using GIPSY6.2 software, and the GPS data processing strategy and estimated parameters were performed according to previous studies [3,8]. The final GPS daily solution was determined according to the International GNSS Service (IGS08) reference frame part of the International Terrestrial Reference Frame (ITRF2008) [18].

GPS time series may be affected by some errors and abnormal steps, such as GPS antenna/instrument replacements and coseismic/postseismic earthquake displacements. We performed the data preprocessing strategy suggested by Zheng et al. [19]. The 3D GPS velocities were estimated in the ITRF 2008 reference frame using the maximum likelihood estimation of CATS software [20].

We incorporated flicker noise plus white noise models as well as annual and semiannual signals when determining the velocity rates and uncertainties. The corresponding results and data of the 3-D velocity and uncertainty in the ETP are presented in Section 3.5.

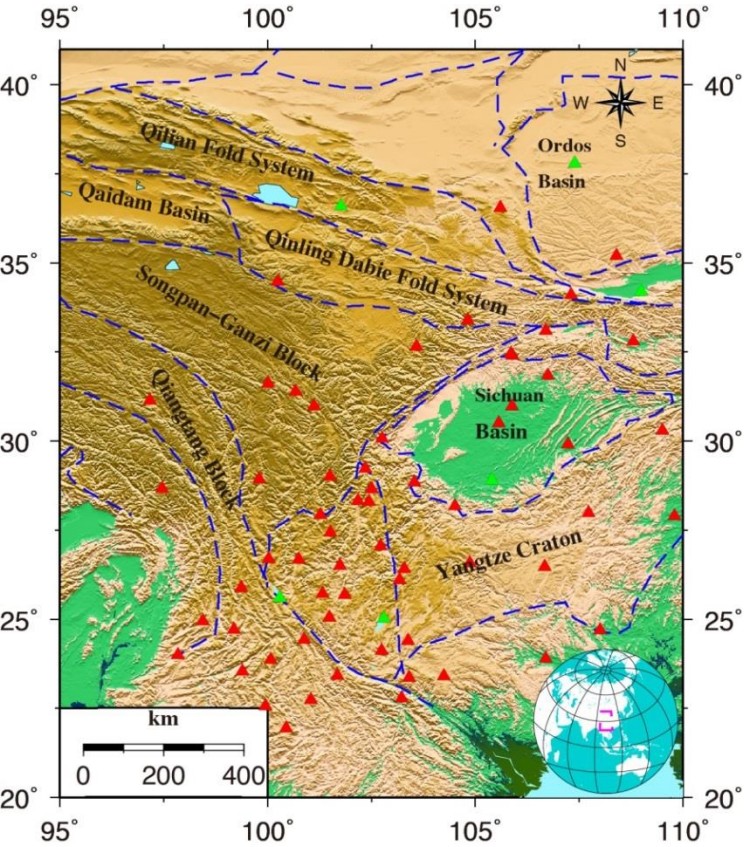

**Figure 1.** Locations of Global Positioning System (GPS) sites from 2000 to 2018 in the Eastern Tibetan Plateau (ETP). The green and red triangles represent GPS time series spanning 1999–2018 and 2010–2018, respectively.

*2.2. GRACE Satellite Gravimetry Data*

The gridded spherical harmonic coefficients of GRACE Release-06 Level-2 (RL06) data products (1° × 1°) for the period from April 2002 to August 2016 were provided by the Center for Space Research at the University of Texas, Austin (CSR) (accessed through ftp://podaac.jpl.nasa.gov/allData/grace/L2/CSR/RL06/). Similar to the RL05 GRACE spherical harmonic coefficient (GSM) products, the spherical harmonic coefficients of RL06 GSM also represent the total gravity variability due to land surface hydrology, cryospheric changes, episodic processes, glacial isostatic adjustment (GIA), and corrections to the background models for atmospheric and oceanic processes. The improvements in the new products of RL06 include improved parameters, processing algorithms, data editing, and background gravity models. More details on the RL06 improvements may be found in the relevant publication of Save et al. 2018 [21].

We applied the harmonic coefficients of GRACE RL06 to compute the surface mass changes of the ETP for the period from April 2002 to August 2016. Before the computation, the coefficients of the C20 term were replaced by satellite laser ranging (SLR) data [22]. Additionally, geocenter motions (the degree-1 coefficients) were estimated, as suggested by Swenson et al. [23]. Then, we used a Gaussian filter with a radius of 300 km and a decorrelation filter (order m=6 with a polynomial of degree 4, P4M6) to eliminate the North–South stripes in GRACE data [24]. The equivalent water height (EWH) can be expressed as [25]:

$$\Delta\sigma(\phi, \theta) = \frac{a\rho_e}{3\rho_w} \sum_{l=0}^{\infty} \sum_{m=0}^{l} \overline{P}_l^m(\cos\theta)\left(\frac{2l+1}{1+k_l}\right)\left[\Delta S_l^m \sin(m\phi) + \Delta C_l^m \cos(m\phi)\right] \tag{1}$$

where $\rho_e$ is the average density of the Earth; $\rho_w$ is the density of water; $a$ is the equatorial radius; $\theta$ and $\phi$ are the colatitude and east longitude, respectively; $\overline{P}_l^m$ is the fully normalized Legendre function of degree n and order m; and $\Delta S_l^m$ and $\Delta C_l^m$ are the monthly Stokes coefficients, respectively.

The remarkable patterns in the EWH distribution in Figure 2 reflect the long-term trend of hydrology in the ETP. We determined that the positive signals throughout the Sichuan and Qaidam Basins are responses to increased precipitation [26,27]. In addition, the negative signals that are present in the Eastern Himalayas are primarily associated with glacial melting [28]. Most of these anomalous phenomena in the ETP, such as hydrologic and glacial variations, are mainly responses to global climate change [14].

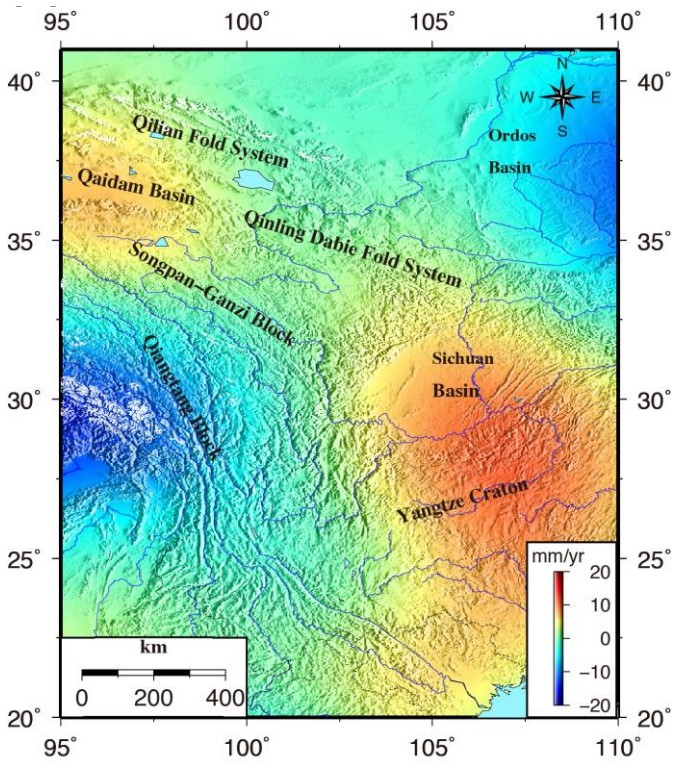

**Figure 2.** Surface mass changes as expressed by the equivalent water height (EWH) derived from the Gravity Recovery and Climate Experiment Release-06 (GRACE RL06) data products throughout the ETP for the period from April 2002 to August 2016.

Furthermore, we used the spherical harmonic coefficients of GSM and GAC (GRACE Average of non-tidal atmosphere and ocean Combination) to model the surface vertical hydrologic, atmospheric and nontidal ocean loading displacements in the ETP. The three-dimensional loading displacement caused by a change in the surface elastic load [3] is calculated as follows:

$$\Delta n = -R \sum_{l=1}^{\infty} \sum_{m=0}^{l} \frac{\partial}{\partial\theta}\overline{P}_{lm}(\cos\theta) \cdot \left[\Delta\overline{C}_{lm}\cos(m\phi) + \Delta\overline{S}_{lm}\sin(m\phi)\right] \cdot \frac{l_l'}{1+k_l'}$$

$$\Delta e = \frac{R}{\sin\theta} \sum_{l=1}^{\infty} \sum_{m=0}^{l} \overline{P}_{lm}(\cos\theta) \cdot m \cdot \left[-\Delta\overline{C}_{lm}\sin(m\phi) + \Delta\overline{S}_{lm}\cos(m\phi)\right] \cdot \frac{l_l'}{1+k_l'} \tag{2}$$

$$\Delta h = R \sum_{l=1}^{\infty} \sum_{m=0}^{l} \overline{P}_{lm}(\cos\theta) \cdot \left[\Delta\overline{C}_{lm}\cos(m\phi) + \Delta\overline{S}_{lm}\sin(m\phi)\right] \cdot \frac{h_l'}{1+k_l'}$$

where $R$ is the Earth's mean radius; $S_{lm}$ and $C_{lm}$ are the spherical harmonic coefficients of the gravity field; and $l'_l$, $h'_l$, and $k'_l$ are the adopted load Love numbers provided by Farrell [29] that are computed relative to the center of mass of the solid Earth [30].

*2.3. Methods*

2.3.1. Principal Component Analysis

The PCA method is an algorithm that decomposes a signal or data set in terms of orthogonal basis functions and can obtain time–domain component series and space–domain spatial fields in geophysics [31]. The detailed algorithm and procedures of the PCA method have been described in previous studies [16,32]. Here, we performed PCA to identify the CMCs of dense GPS time series throughout the ETP.

2.3.2. Wavelet Spectrum Analysis

We used the wavelet spectrum to reveal the oscillatory signals contained in each CMC's time series as well as the GRACE-modeled surface loading displacements, primarily on seasonal-to-interannual timescales. The corresponding wavelet spectrum of a time series $f(t)$ is calculated as follows [33,34]:

$$F(a,b) = \frac{1}{\sqrt{a}} \int_{-\infty}^{\infty} f(t) W\left(\frac{t-b}{a}\right) dt \qquad (3)$$

where $W\left(\frac{t-b}{a}\right) = \frac{1}{\sqrt{2a\pi}} e^{-\frac{(t-b)^2}{2a^2}} \cos\left(\pi \sqrt{\frac{2}{\ln 2}} \frac{(t-b)}{a}\right)$ is the base wavelet with an effective length shorter than the target time series $f(t)$; $a$ and $b$ are the variables corresponding to the dilation/compression scale factor and the "sliding" of the wavelet over $f(t)$, respectively. To maintain the polarity of the wavelet transform (the positive/negative phases of the undulations), the real-valued Morlet function is employed for $W(t)$. $F(a,b)$ is related to the time-frequency domain of signals that includes the frequencies of the signals and the times associated with those frequencies. The details of $F(a,b)$ related to how the wavelet function is processed are as follows:

- Set a matrix size $S$ as $\begin{pmatrix} S_{11} & \cdots & S_{1j} \\ \vdots & \ddots & \vdots \\ S_{i1} & \cdots & S_{ij} \end{pmatrix}$, where i is the number of periods/frequencies we need in the transform and j is the length of the time series data;

- Determine the data range and interval of $t$ in $W\left(\frac{t}{a}\right)$;

- Estimate the convolution of $f(t)$ and $W\left(\frac{t}{a}\right)$;

- Adjust the time shift factor of the data referring to the value of b in $W\left(\frac{t-b}{a}\right)$ and multiply it by a coefficient of $\frac{1}{\sqrt{a}}$ to obtain $F(a,b)$;

- Insert the results into the corresponding matrix $S$ in step (1);

- Finally, two results are obtained, which include the matrix $S$ induced by wavelet transform and the periodic series of $P_i$.

## 3. Results

*3.1. The Spatial-Temporal Patterns of CMCs*

The PCA method was used to decompose the 69 GPS time series throughout the ETP. Here, seasonal signals (including annual and semiannual signals) and long-term trends were removed from each GPS time series before applying the PCA. We investigate the six long-term GPS time series for the period from 1999 to 2017 and the 63 short-term GPS time series for the period from 2010 to 2017 separately. The spatial patterns of the first three CMCs of the long-term and short-term GPS time series

representing the normalized amplitudes of the eigenvectors (the scaled principal components PC1, PC2, and PC3) are presented in Figure 3. For the six long-term GPS stations, the average response of spatial eigenvectors for PC1 on the North, East, and vertical components are 58%, 37%, and 46%. The average response of spatial eigenvectors for PC2 on the North, East, and vertical components are 36%, −60%, and 65%, and for PC3, they are −59%, −39%, and −38%, respectively. In addition, for the 63 short-term GPS stations, the average response of spatial eigenvectors for PC1 on the North, East, and vertical components are 60%, 50%, and 51%. The average response of spatial eigenvectors for PC2 on the North, East, and vertical components are −24%, 25%, and 43%, and for PC3, they are 20%, 22%, and −25%, respectively. The average response of spatial eigenvectors reflect which station above (or below) the average.

Histograms of the eigenvalues sorted from low to high PC orders on the North, East, and vertical components are presented in Figure 4. The PC eigenvalues gradually decrease from low to high PC orders. Low orders have a common long-wavelength pattern, while high orders are mixed with local effects and other noises (namely, the long-period signals are mainly contained in the lower-order PCs, and vice versa) [35]. Eigenvectors are arranged according to the data size, which consist with the powers of eigenvalues from low to high orders. Generally, the long-period signals will affect the estimations of the velocities from the GPS time series, so we need to employ only the lower-order PCs. In this study, we only investigate the first three PCs shown in Figure 4.

PC1, PC2, and PC3 on the North, East, and vertical components for the long-term GPS time series are shown in Figure 5a,c,e, respectively; their corresponding Fourier amplitude spectra are shown in Figure 5b,d,f, respectively. The similar results for the first three PCs from the short-term GPS time series are shown in Figure 6. The Fourier spectra in Figures 5 and 6 show that the strongest long-period signals are mainly contained in PC1, and some residual long-period signals are still present in PC2 and PC3. As the first three PCs contain almost all the significant signals that have a period longer than 1 yr, we finally decide to use the sum of PC1, PC2, and PC3 as the CMC in this study.

Here, we note that the look-like 'annual' signal in Figure 5c is just the envelope of some high frequency noise, not the real annual signal. This can be easily validated by the filter process (we will not show this further). As for the spectral peak in Figure 5f around annual period, the frequency of it is 1.048 cpy. This signal is the so-called "GPS year" at 1.04 cpy, which is related to artifact that can be traced to certain orbit modeling defects or aliasing of site-dependent positioning biases modulated by the varying satellite orbit geometry [36]. As we only fitted and removed the signal with a 1.0 cpy frequency, this signal is still presented in the residual time series.

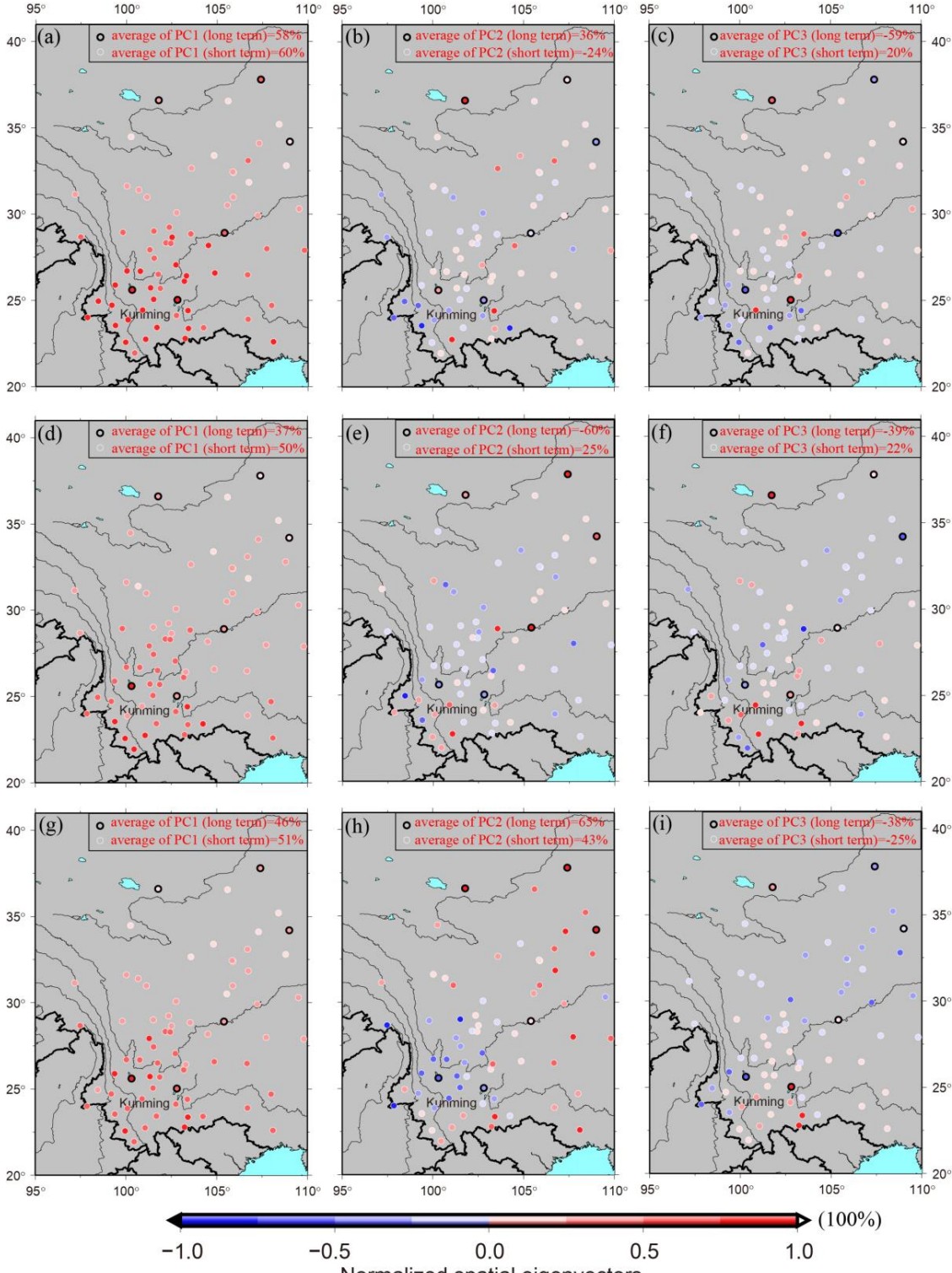

**Figure 3.** Spatial patterns of the scaled principle components (PCs) in the ETP. Red symbols represent positive responses to the scaled PC, whereas blue symbols represent negative responses to the scaled PC. The symbols with black circles denote 6 long-term GPS sites, while those with white circles denote 63 short-term GPS sites. (**a–c**) are the normalized spatial eigenvectors from PC1 to PC3 on the North component; (**d–f**) are the normalized spatial eigenvectors from PC1 to PC3 on the East component; and (**g–i**) are the normalized spatial eigenvectors from PC1 to PC3 on the vertical component. The scaled dots with lengths ranging from −1 to 1 indicate a normalized range from −100% to 100%.

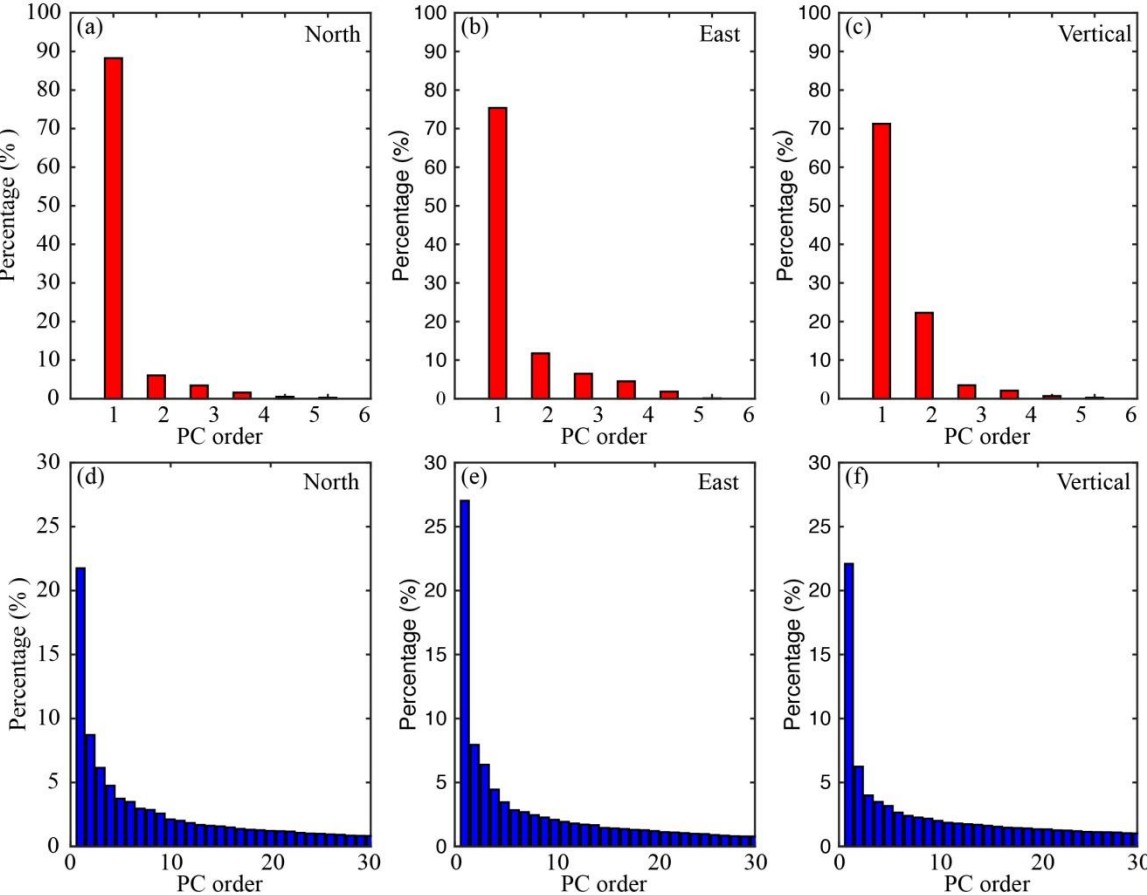

**Figure 4.** Histograms of the eigenvalues sorted according to the PC order on the three GPS components (North, East, and vertical). The top three histograms are the percentages of the PC eigenvalues of GPS long-term observations, while the bottom three are the percentages of the PC eigenvalues of GPS short-term observations.

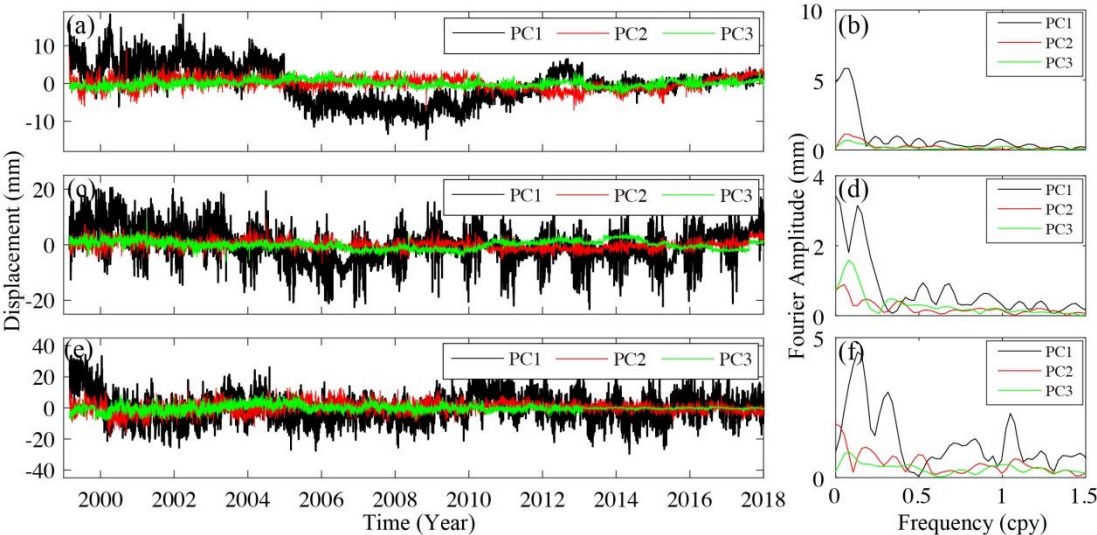

**Figure 5.** Time series of the common mode components (scaled PC1 through PC3) decomposed by principal component analysis (PCA) from the regional long-term GPS time series. (**a**–**f**) are the PCs on the North, East, and vertical components, and the corresponding Fourier amplitude spectra of the first three PCs on the North, East, and vertical components, respectively.

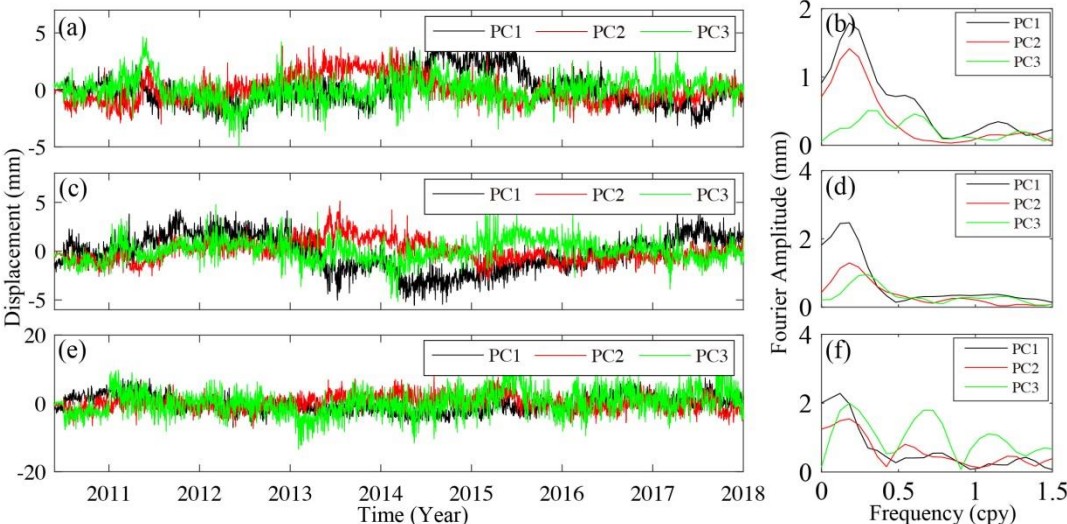

**Figure 6.** Time series of the common mode components (scaled PC1 through PC3) decomposed by principal component analysis (PCA) from the regional short-term GPS time series. (**a**–**f**) are the same as those in Figure 5.

### 3.2. Wavelet and Fourier Spectra of Common Mode Components

As mentioned above, we added PC1, PC2, and PC3 into one time series to represent the CME (see Figure 7a) and then used the wavelet and Fourier spectra to analyze the CME. Figure 7b–g show the corresponding wavelet and Fourier spectra for the long-term GPS CME time series, while Figure 8 shows the same spectra as Figure 7, but for the short-term GPS time series. Figure 7f shows an oscillation with a period of approximately one year. As we have removed the signal with a period of one yr from the GPS time series by using the least squares method before employing PCA, and Figure 7g shows that the frequency of this signal is approximately 1.03 cpy, we suggest that a potential interpretation of this frequency is attributed to the GPS draconic error related to the satellite orbit [36–39]. Figure 7f also clearly shows a signal with a period of ~6 yr, and the corresponding Fourier spectrum in Figure 7e also confirms this finding. However, in Figure 7c,e, there is no isolated spectral peak for this signal (see the vertical dashed lines), and the wavelet spectra in Figure 7b,d further shows that some longer period (>6 yr) signals overlap each other (the length of the used record is not long enough to have enough frequency resolution, so the spectral peaks (or lines in wavelet spectrum) of the longer period signals cannot be isolated from that of the ~6 yr signal). Given that there are many normal modes of the Earth that have periods longer than 10 yr [40,41], signals with such periods probably be recorded by the GPS time series; consequently, the extra-low frequency terms in the GPS time series will be affected by them (this is just a suggestion, the more details should be further studied in the future). We compared our ~6 yr signal with the results from Ding and Chao (2018b) [42], who first detected a signal with a period of ~6 yr from global GPS observations based on a stacking method, and we found that the phase of the ~6 yr signal in this study is almost the same as that of the signal in Ding and Chao (2018b) [42]. Furthermore, if we accept the explanation of Ding and Chao (2018b) [42], namely, that the source of this ~6 yr signal is mantle-inner core gravity coupling (MICG), we can calculate that the theoretical amplitudes for Tibet are approximately 1.2 mm, while Figure 7e shows that the amplitude of the common ~6 yr signal is approximately 1.3 mm; if we consider noise, the theoretical amplitude is almost the same as the observed value. Given these findings, we tend to believe that the MICG suggested by Ding and Chao (2018b) [42] is the mechanism for the ~6 yr signal in the CMC. Additionally, according to Ding and Chao (2018b) [42], the ~6 yr signals on the North and East components for the Tibet area have amplitudes of approximately 0.2 mm and 0.08 mm, respectively, which are close to the noise levels shown in Figure 7b,d. This can also explain why the ~6 yr signal cannot be found in the CMCs on the North and East components.

As for the results from the short-term GPS time series, only the vertical component clearly reveals the ~6 yr signal (see the vertical dashed lines in Figure 8). The most significant spectral peaks in Figure 8a,b still contain some overlapping signals (see the gray shaded areas). The ~6 yr signal obtained from the short-term GPS time series has the same phase as that from the long-term GPS time series, which means that the ~6 yr signal is almost stable and periodic, which is consistent with the conclusion of Ding and Chao (2018b) [42]. In addition, the amplitude of the ~6 yr signal in Figure 8g is ~2.4 mm, but the wavelet spectrum in Figure 8f shows that the ~6 yr signal is affected by signals with a period of approximately 2–5 yr; therefore, this spectral amplitude cannot represent the real amplitude of the ~6 yr signal. Considering that the background noise level is approximately 1 mm (see Figure 8g), an amplitude of ~2.4 mm is also acceptable.

In addition to the frequency bands discussed above, some ~2–5 yr signals are also present in the wavelet spectrum in Figure 7f (especially the time span denoted by the dashed ellipse) that are likely caused by El Niño-Southern Oscillation (ENSO) meteorological oscillations (see Section 3.4). To further confirm this, we will further investigate the surface elastic displacements caused by hydrologic, atmospheric, and nontidal ocean loads (Section 3.3) and the ENSO index (Section 3.4) to determine whether relations exist between them and the observed signals.

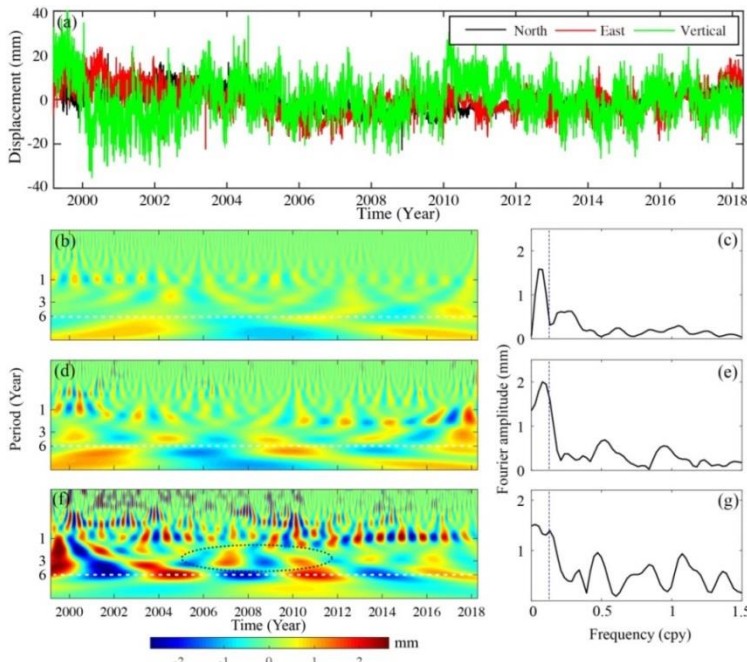

**Figure 7.** Spectrum analysis of common mode components in the long-term GPS time series. (**a**) shows the time series sum of PC1, PC2, and PC3. (**b**) and (**c**) are the wavelet time-frequency spectra and Fourier spectrum, respectively, on the North component. (**d**) and (**e**) are the wavelet time-frequency spectra and Fourier spectrum, respectively, on the East component. (**f**) and (**g**) are the wavelet time-frequency spectra and Fourier spectrum, respectively, on the vertical component. The ellipse in (**f**) denotes the possible ENSO (El Niño-Southern Oscillation) effect; and the vertical dashed lines in (**c**), (**e**), and (**g**) denote the ~6 yr signal.

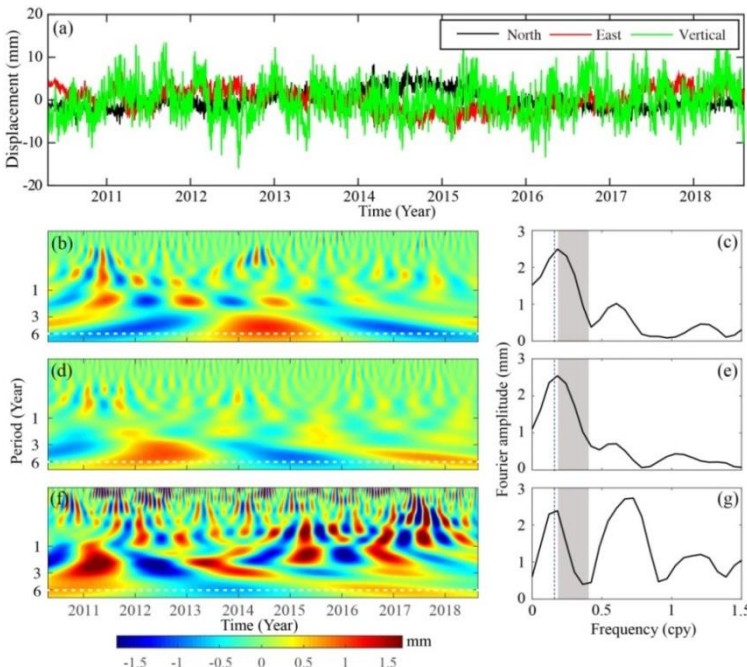

**Figure 8.** Spectrum analysis of common mode components in the short-term GPS time series. (**a**)–(**g**) are the same as those in Figure 7. The gray areas in (**c**), (**e**), and (**g**) are used to compare the different spectral peaks around the 3–6yr (the vertical dashed lines), comparing with the spectral peak in (**g**), the corresponding spectral peaks in (**c**) and (**e**) are clearly wider. This means that there are some other signals presented in the 3–6 yr period band in the Northern and Eastern components. The wavelet spectra in (**b**), (**d**), and (**f**) also confirm this result.

### 3.3. Surface Elastic Deformation Due to Changing Loads

Variations in surface loads due to the Earth's fluid envelope deform the elastic lithosphere. This deformation is captured by GPS, especially in the vertical component. A potential geophysical interpretation of the PC series is that large-scale surface elastic loads are a result of hydrologic, atmospheric and nontidal ocean variations. Therefore, we calculate surface elastic loading displacements according to Formula (2) using GRACE RL06 gravity data, including the spherical harmonic coefficients of the GRACE solution (GSM) and the GAC.

We averaged the GRACE-inferred (detrended) loading time series corresponding to the locations of the 69 GPS sites to reveal the regional seasonal and interannual oscillations of the hydrologic (GSM-derived), atmospheric and nontidal ocean (GAC-derived) loading displacements in the ETP. The horizontal and vertical loading displacements are presented in Figures 9 and 10, respectively. We removed the seasonal (annual and semiannual) signals from the loading displacements using the least squares method and then applied the Morlet wavelet spectrum to identify harmonic interannual signals in the elastic loading displacement.

We found significant interannual oscillations with a period of ~2–5 yr; however, the phase and amplitude show poor consistency with the CMCs on the horizontal components between the GPS and GRACE data. While previous studies [43,44] also show that horizontal displacement amplitudes from GRACE are systematically underpredicted and out of phase with GPS station time series, they also explained that such discrepancies can be mostly reconciled by using a degree-1 contribution derived from the GPS dataset. Here, we just note this without further study, additional studies can be done to further validate such findings in the future.

On the vertical component, as shown in Figure 9k,l, signals corresponding to hydrologic loading (the GSM-derived solutions) in the ~2–5 yr frequency band are evident; however, these signals do not include some patterns observed in the atmospheric and nontidal ocean loading displacements. The

wavelet spectra between 2005 and 2012 show consistent patterns with Figure 7(f), indicating that the water mass balance deforms the crust at the surface. The presented interannual oscillations could be used to interpret the vertical CMC component.

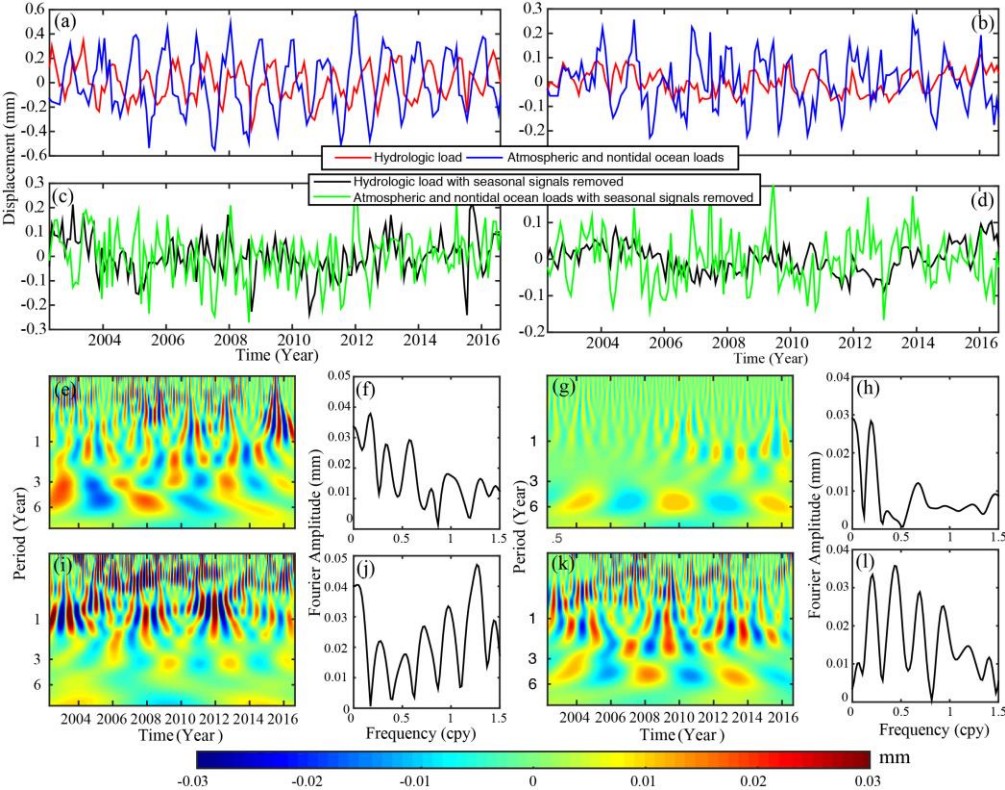

**Figure 9.** Horizontal seasonal and interannual signals of atmospheric, hydrologic, and nontidal ocean loading displacements from GRACE (Gravity Recovery and Climate Experiment) modeled mass changes corresponding to the locations of GPS sites in the ETP. (**a**) and (**b**) are the average GRACE-modeled elastic displacements resulting from hydrologic (GSM), atmospheric, and nontidal ocean (GAC) loads on the North and East components, respectively, and (**c**) and (**d**) are the corresponding elastic displacements with the seasonal (annual and semiannual) signals removed. (**e**) and (**f**) are the wavelet time-frequency spectra and Fourier spectrum of GAC (with seasonal signals removed) on the North component. (**g**) and (**h**) are the wavelet time-frequency spectra and Fourier spectrum of GAC (with seasonal signals removed) on the East component. (**i**) and (**j**) are the wavelet time–frequency spectra and Fourier spectrum of GSM (with seasonal signals removed) on the North component. (**k**) and (**l**) are the wavelet time–frequency spectra and Fourier spectrum of GSM (with seasonal signals removed) on the East component.

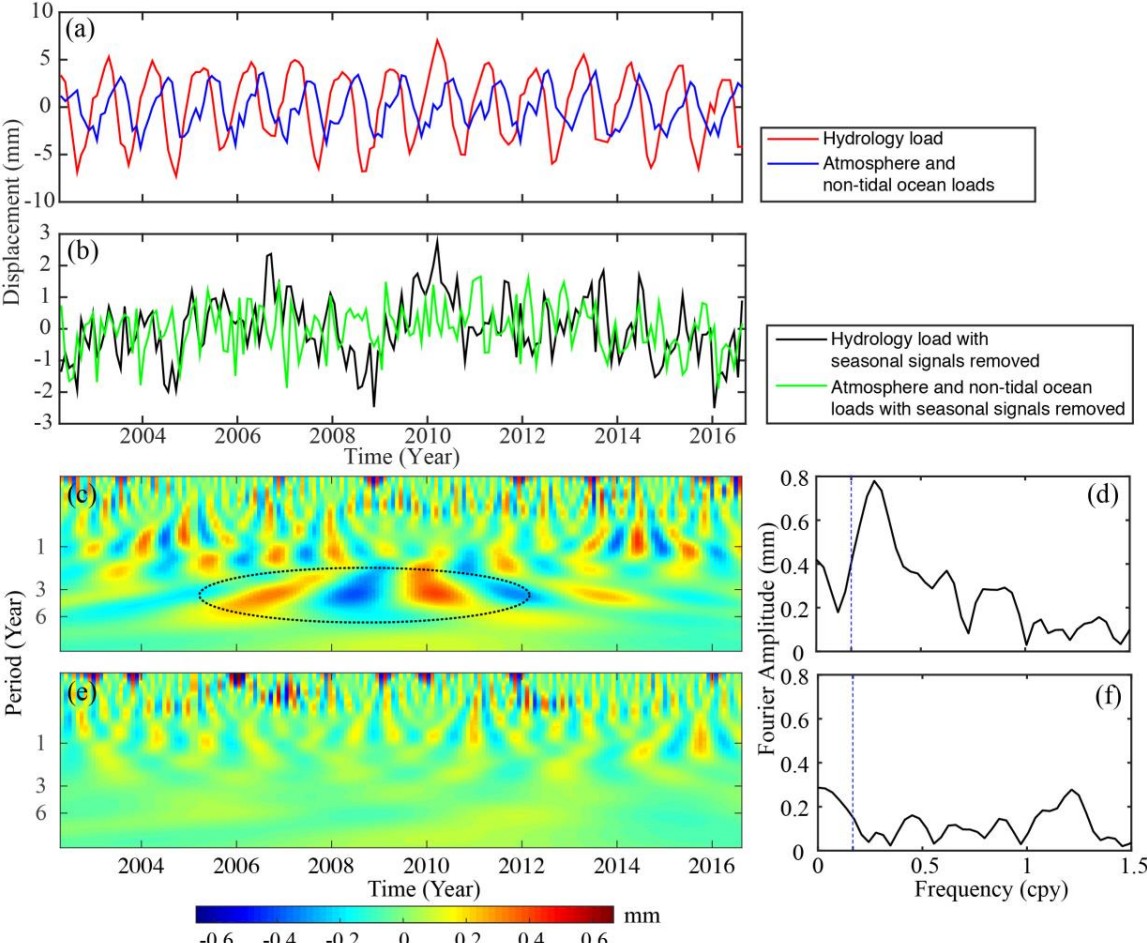

**Figure 10.** Vertical seasonal and interannual signals of atmospheric, hydrologic, and nontidal ocean loading displacements from GRACE-modeled mass changes corresponding to the locations of GPS sites in the ETP. (**a**) shows the average GRACE-modeled elastic displacements of hydrologic (GSM), atmospheric, and nontidal ocean (GAC) loads and (**b**) shows the corresponding elastic displacements with seasonal (annual and semiannual) signals removed (bottom). (**c**) and (**d**) are the wavelet time-frequency spectra and Fourier spectrum of GSM (seasonal signals removed). (**e**) and (**f**) are the wavelet time-frequency spectra and Fourier spectrum of GAC (seasonal signals removed).

### 3.4. ENSO Pattern Analysis

Furthermore, we calculated a climate response index, the Southern Oscillation Index (SOI), with the wavelet time-frequency spectra, as shown in Figure 11. The SOI, which is well known to exhibit a periodic fluctuation (i.e., every 2–5 years) [45], as shown in Figure 11b, is a measure of the large-scale fluctuations in air pressure occurring between the Western and Eastern tropical Pacific. The interannual dynamics of the SOI are mainly related to global large-scale variations in precipitation and temperature that deform the Earth's surface. The SOI wavelet spectrum presents a significant ~3 year oscillation, which reveals ENSO events during 2005–2012. Simultaneously, the CMCs and hydrologic loading deformation shows anomalous oscillations during this period, which are a response to global climate change. Therefore, global climate change also has a potential effect on regional hydrological loading that is manifested by interannual oscillation of CMCs.

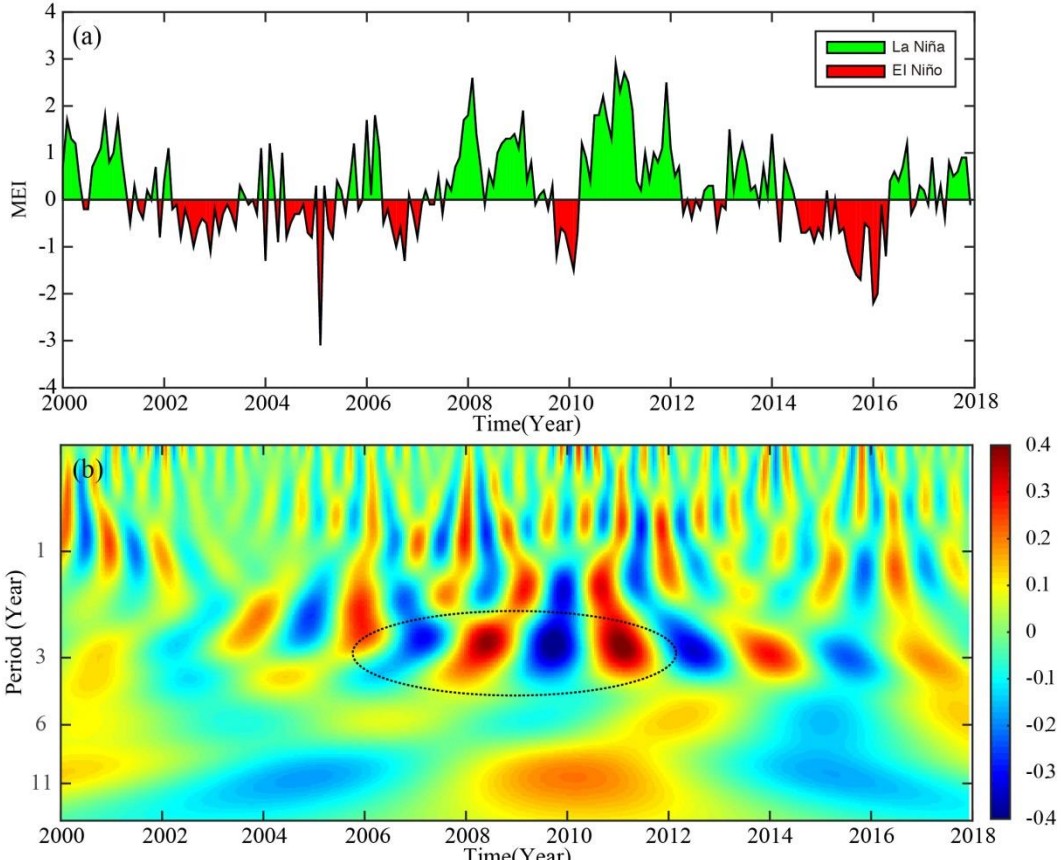

**Figure 11.** (**a**) Southern Oscillation Index (SOI) during the period from 2000 to 2018. Prolonged periods of positive (negative) SOI values coincide with La Niña (El Niño) episodes. (**b**) Corresponding wavelet time–frequency spectra of the SOI.

### 3.5. Three-dimensional Crustal Velocity of the ETP

After correcting the GPS time series for the CMCs, we applied the MLE (maximum likelihood estimation) approach, as described by previous studies [20,46], to analyze the GPS velocities and uncertainties. The velocity uncertainties on the three components over the ETP are presented in Figure 12. The values of these uncertainties decreased after filtering out the CMCs, especially those on the vertical component. The average differences between the unfiltered and filtered CMCs were 0.02, 0.02, and 0.10 mm/yr for the North, East, and vertical velocity uncertainties, respectively. The velocity uncertainties that gradually decreased with the longer observation time spans of the GPS time series indicate that errors in GPS velocities are likely dominated by biases resulting from multipath errors and unmodeled nonsecular Earth deformations.

Figure 13 presents the horizontal and vertical residual velocities, which reveal the effect of CMC on the GPS three-dimensional velocity in the ETP. The horizontal residual velocity is very small, with a maximum of ~0.47 mm/yr for the GPS site of SCGY (105.8°E, 32.4°N). Compared with the horizontal velocity of crustal deformation occurring at a rate of ~15–20 mm/yr, as shown in Figure 14, this residual value appears to have no significant influence on the final horizontal velocity. However, the influence on the vertical velocity was different, although the value varied between approximately −0.6 and 0.8 mm/yr. Vertical crustal deformation, including nontectonic and tectonic deformation, primarily varies between −3 and 3 mm/yr in the Tibetan Plateau [3,4]. Therefore, the unmodeled CMCs contribute significantly to the vertical velocity, which should be considered in the estimation of vertical crustal deformation.

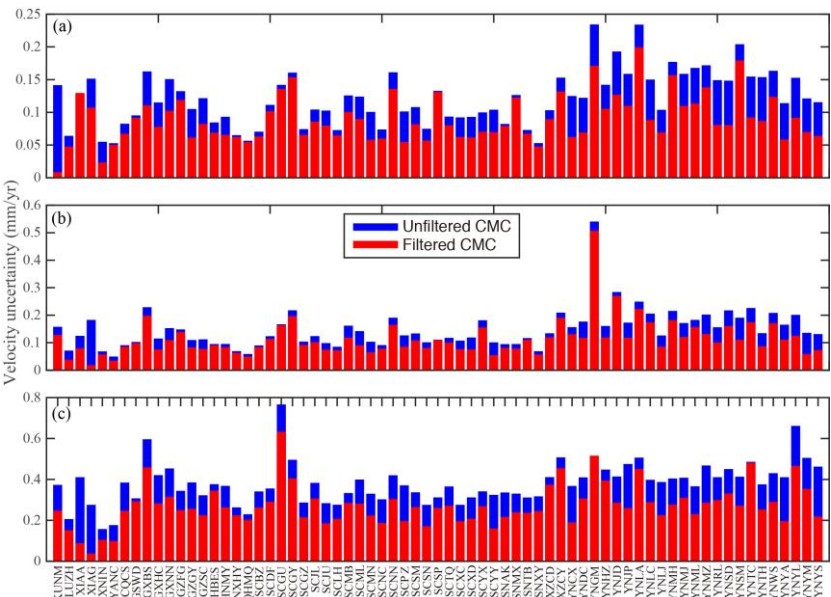

**Figure 12.** Velocity uncertainties on the three components (North, East, and vertical) with unfiltered and filtered common mode errors in the GPS time series. The red and blue lines are the averaged velocity uncertainties of the unfiltered and filtered common mode components (CMCs), respectively.

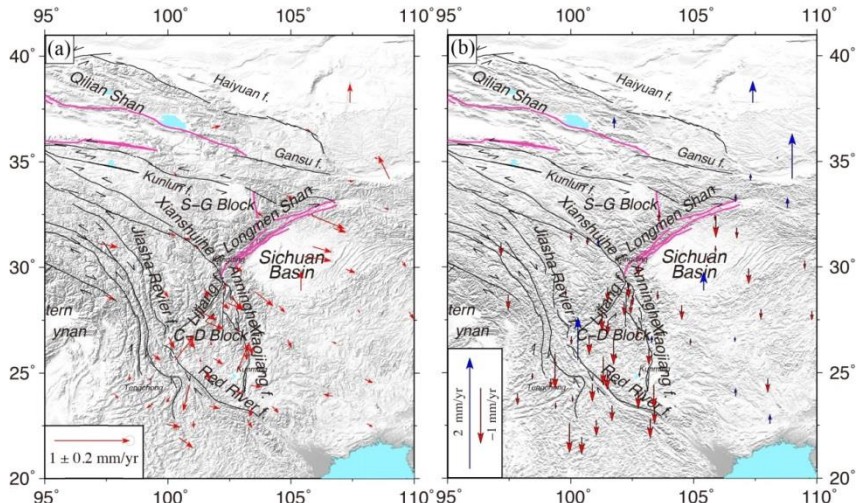

**Figure 13.** Residual effect on the three-dimensional GPS velocity in the ETP. (**a**) Residual of the difference between the unfiltered and filtered CMCs of the horizontal GPS velocity with respect to the ITRF 2008 frame; (**b**) residual of the vertical velocity with respect to the ITRF 2008 frame.

Removing the CME from the original GPS time series is equal to the application of a regional filter [16,35]. To identify the three-dimensional crustal deformation throughout the ETP, we converted the velocity vectors into a Eurasia-fixed reference frame based on the IGS-Eurasia Euler vector ($w_x = -0.0247$, $w_y = -0.1418$, and $w_z = 0.2093° / Myr$) provided by Kreemer et al. [47]. The three-dimensional velocity of ground deformation in the ETP is presented in Figure 14. The 3-D velocity reflects the kinematic crustal deformation in the ETP, which is consistent with the findings of previous research [4]. The GPS-derived horizontal velocity reveals Eastward crustal extrusion in the central Tibetan Plateau at a rate of ~15–20 mm/yr relative to the neighboring Sichuan Basin [6].

The stable Eurasia reference frame-based horizontal crustal deformation of the ETP can be summarized as follows:

- There is crustal clockwise rotation around the eastern end of the Himalayan syntaxis.

- There are movement differences among blocks, for example, ESE lateral crustal extension along the Chuan-Dian block and NNE shortening in the upper part of the Songpan-Ganzi block.
- Shortening accommodates the Longmen Shan faults due to the resistance of the Yangtze Craton block, and the postseismic deformation along the Longmen Shan produces anomalous vertical crustal deformation that leads to uplift.

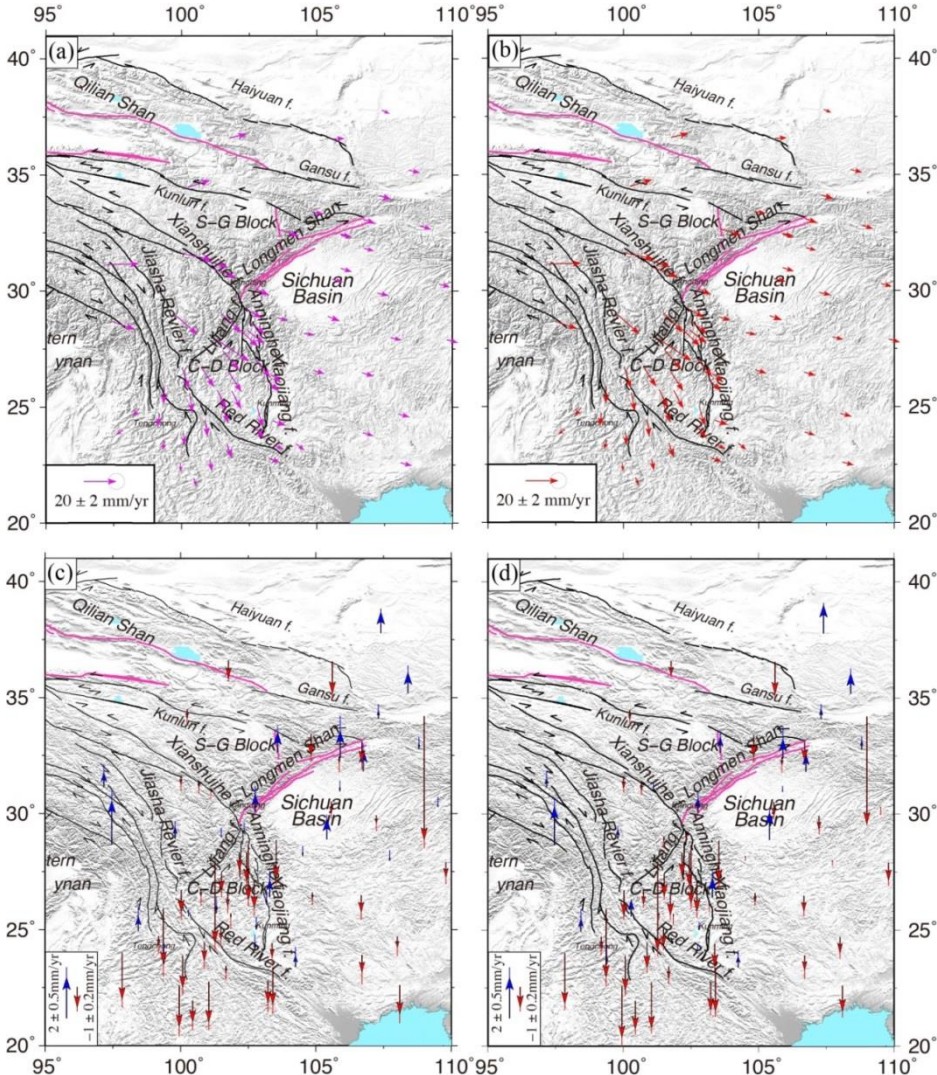

**Figure 14.** Three-dimensional crustal deformation in the ETP. (**a**) and (**b**) are the horizontal GPS velocities with respect to the stable Eurasian reference frame with an unfiltered CMC and a filtered CMC, respectively. (**c**) and (**d**) are the vertical velocities with respect to the ITRF 2008 frame with an unfiltered CMC and a filtered CMC, respectively.

Similar to previous studies, a more detailed interpretation of this GPS-derived horizontal crustal deformation is not provided. Comparing the crustal deformation revealed herein with that reported in previous studies [3,4], the primary difference in our results concerns the vertical velocity, especially in the Sichuan–Yunnan area. In this study, we did not consider the effect of surface mass elastic deformation, which contributes to nontectonic motion. The nontectonic elastic displacement derived from GRACE data in the ETP ranges from approximately 0.1 mm/yr to 0.4 mm/yr, and the changes are primarily concentrated in the Sichuan Basin due to increased precipitation [3]. The elastic deformation in southern Yunnan is less than 0.1 mm/yr, which has limited influences on vertical tectonic deformation.

Therefore, the sinking crustal deformation in Southern Yuannan is primarily a response to crustal extension and to flow loss in the lower crust.

## 4. Discussion

### 4.1. Nonlinear Signals of an Unmodeled CMC

GPS time series may contain seasonal, interannual, unmodeled signals, and noises, which may be explained by functional and stochastic noise models [39]. Therefore, a stochastic model that accounts for temporal correlations in the GPS time series should be incorporated to identify realistic geophysical information. Unprocessed CMCs result in long-period noise within the GPS coordinated time series, which may bias subsequent velocity estimates and increase velocity uncertainties [48]. Some spatial filtering methods have been proposed to diminish the impacts of CMCs existing in regional or global dense GPS networks [16,49,50].

We applied the Morlet wavelet spectrum and Fourier spectrum to identify the characteristic patterns of the interannual CMC signals decomposed from dense GPS sites in the ETP, following which we identified evident signals with periods of ~2–5 yr, as shown in Figures 7 and 8. The GRACE-modeled surface loads also present the interannual oscillations, which are a response to the redistribution of elastic Earth's surface mass (including hydrologic, atmospheric, and nontidal ocean loading) (Figures 9 and 10). The vertical component shows good consistency in the frequency band of ~2–5 yr, while the amplitude and phase on the horizontal components are both inconsistent between the GPS and GRACE data. In addition, we found that the periods and phases of the ~6 yr signal obtained from the vertical CMCs of the long-term and short-term GPS time series are almost the same as each other. More importantly, our results are also consistent with the inferred ~6 yr signal from the global GPS observations by Ding and Chao (2018b) [43]; this is true not only for the phase, but also for the amplitude. Given these findings, we tend to agree with the suggested physical mechanism for the ~6 yr signal proposed by Ding and Chao (2018b) [43], namely, that MICG causes a rotary westward motion with a period of ~6 yr in the Earth's outer core, and the induced pressure deforms the core-mantle boundary and further produces displacements at the Earth's surface that can be recorded by GPS networks.

### 4.2. Velocities and Uncertainties of the ETP

GPS-derived velocities may be biased by modeled seasonal signals and noise, especially on the vertical component, with an observation time of less than 2.5 years [51]. Moreover, unmodeled nonstationary signals will also have an effect on velocities and their uncertainties [35,52]. Filtering unmodeled CMCs will not only improve the precision of the velocity due to the interannual elastic loading correction, but also improve the signal-to-noise ratio of the velocity [16,53]. Therefore, the CMCs in GPS time series should be considered when estimating the ground velocity using regionally dense GPS sites. The velocity uncertainty is significantly improved when CMCs are filtered from GPS time series; this is especially true for the vertical velocity in the ETP, as shown in Figure 10. The differences in the velocity vectors between the unfiltered and filtered CMCs averaged 0.06 and 0.2 mm/yr for the horizontal (based on the Eurasian reference frame) and vertical (based on the ITRF2008 reference frame) velocities, respectively. The improvement in the velocity precision was more significant for the vertical crustal deformation after removing the CMCs, especially for the vertical crustal motion of the ETP.

## 5. Conclusions

In this study, we used 69 continuous GPS sites, as shown in Figure 1, to identify the surface seasonal and interannual elastic deformations and three-dimensional crustal deformation of the ETP. The interannual oscillation signals in geodetic measurements were highlighted in this study. PCA and wavelet methods were used to perform time series analysis of the PCs decomposed from a dense

GPS time series and GRACE-inferred load displacements to reveal the spatially varying patterns and nonlinear signals of the common mode components in the GPS time series. We found that the patterns of the residual time series in the vertical component are evidently mixed with annual and interannual signals. The residual annual signal is most likely related to the draconitic period (~1 cpy), which is induced by the satellite orbit. Interannual signals with a period of ~2–5 years were also identified. We further discovered that the GRACE-derived elastic loading displacements present similar patterns represented by the quasi-periodic oscillation of La Niña (El Niño) phenomena that may reflect responses to global climate change. Furthermore, a signal was identified in the CMCs; this signal might be related to the oscillations with a period of ~6 yr in the displacements induced by a rotary westward motion with a period of ~6 yr in the Earth's outer core. These results reveal that environmental loading displacements and geodynamics are excitation sources of interannual CMC signals. We compared the difference in velocities with unfiltered and filtered CMCs and confirmed that the velocity uncertainties decrease by 0.2 mm/yr on the vertical component. Finally, the contemporary 3-D velocity was presented to interpret the crustal deformation and dynamic processes of the ETP.

**Author Contributions:** All authors contributed significantly to the manuscript. Y.P. performed all data processing and analyses and contributed to the manuscript draft. R.C. is the main author and initiated the idea, provided critical comments and contributed to the final revision of the paper. H.D., X.X. and W.S. provided critical comments about and modification to the manuscript. G.Z. modified the paper format and performed GPS data processing. Y.X. and S.L. provided technical guidance regarding GRACE data processing.

**Funding:** This research was funded by the NSFC (grant Nos. 41904012, 41774024, 41974022), China Postdoctoral Science Foundation (No. 2018M630879), the Open Research Fund Program of the Key Laboratory of Geospace Environment and Geodesy, Ministry of Education, China (No. 17-02-06) and Guangxi Key Laboratory of Spatial Information and Geomatics, China (No. 16-380-25-32).

**Acknowledgments:** The GPS data used in this paper are primarily from the National Key Scientific Projects "Tectonic and Environmental Observation Network of Mainland China" (CMONOC I and II) and Caltech Tectonics Observatory. We are grateful to IGS for providing global GPS observations and products and to JPL for providing the GIPSY6.2 software.

**Conflicts of Interest:** The authors declare no conflict of interest.

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
