# Peer review of "Common Mode Component and Its Potential Effect on GPS-Inferred Three-Dimensional Crustal Deformations in the Eastern Tibetan Plateau"

_remotesensing, doi:10.3390/rs11171975_

Round 1
Reviewer 1 Report
This revised manuscript is improved significantly. Most of my comments are considered and implemented. However, there are still a few weaknesses in this version. I think this version is publishable and hope that these weaknesses can be remedied in the final version. The followings are my main concerns:
1、 Section 3.1: Please provide clear statement about the purpose and function of calculating the “average normalized spatial eigenvectors”. The positive or negative, large or small “average normalized spatial eigenvectors” stand for what?
2、 Figure 5: PC1 of east component (c) displays clear annual (or draconic period) modulation of high frequent oscillation. But its frequency spectrum (d) does not match the time domain behavior. Meanwhile the spectrum (f) shows significant annual peak. How to explain this? What is the nature of such a modulation? This revised manuscript exhibits a lot of analyses and pictures, but is lack of corresponding explanations about the mechanism and nature of the derived results.
3、 Line 241: “Given that there are many normal modes of the Earth that have periods longer than 10 yr [40,41], signals with such periods could also be present in the GPS time series; consequently, the highest spectral peaks in Figures 7a and 7b and the first significant spectral peaks in Figure 7c might be caused by such overlapping long-period signals.” This bold statement is likely speculate. First, Earth’s global mode is usually absorbed by 7-parameter transformation during the reference frame alignment. Hence it is hard to be left in the site coordinate time series. Second, low frequency systematic errors cannot be excluded.
4、 Figure 7 and Figure 10, pattern inside the dash circle: Both spectrum show 2.5 years quasi-oscillation. But the wavelet analyses indicate negative correlation. How to explain this?
Author Response
Response to Reviewer #1
This revised manuscript is improved significantly. Most of my comments are considered and implemented. However, there are still a few weaknesses in this version. I think this version is publishable and hope that these weaknesses can be remedied in the final version. The followings are my main concerns:
Response: Thanks for your thoughtful suggestions and comments. We have improved our manuscript as you suggested, see in revised version.
1、Section 3.1: Please provide clear statement about the purpose and function of calculating the “average normalized spatial eigenvectors”. The positive or negative, large or small “average normalized spatial eigenvectors” stand for what?
Response: Thanks for your significant suggestion. In fact, there is no specific meaning, similar to our usual indicators for the maximum, minimum and average. After giving the average value, you can see which ones stand above the average (below). We have added relate statement in revised manuscript.
2、Figure 5: PC1 of east component (c) displays clear annual (or draconic period) modulation of high frequent oscillation. But its frequency spectrum (d) does not match the time domain behavior. Meanwhile the spectrum (f) shows significant annual peak. How to explain this? What is the nature of such a modulation? This revised manuscript exhibits a lot of analyses and pictures, but is lack of corresponding explanations about the mechanism and nature of the derived results.
Response: Thanks for your careful review. As for the look-like annual signal in (c), it is actually caused by some high-frequency noise. Please see the following Figure R1a, the PC1 and its filtered time series are plotted. Here we use a low-pass filter (signal with f>1.5cpy with be filtered). R1c shows the Fourier spectra of these two time series in R1a, we can see that the spectral peaks for the annual signal are not change before and after the filter process, while the look-like ‘annual signal’ is not presented in the filtered time series any more. Figure R1c shows a segment of the Figure R1a to more clearly show that the look-like ‘annual signal’ just some high frequency noise. As the reason of such high frequency noise, we have no more explanation, they may be caused by the station itself.
According to your comments, we add some explanations, please see Lines 203-204.
Figure R1. (a) The PC1 shown in Figure 5c and its filtered time series; (b) the Fourier spectra of the two time series in (a); (c) a segment of (a).
As for the spectral peak in (f) around annual period, the frequency of it is 1.048cpy. This signal is the so-called “GPS year” at 1.04 cpy, which is related to artifact that can be traced to certain orbit modeling defects or aliasing of site-dependent positioning biases modulated by the varying satellite orbit geometry (Ray et al., 2008). As we only fitted and removed the signal with a 1.0cpy frequency, so this signal is still presented in the residual time series. According to your comments, we add an explanation about this signal, please see line 210-215.
3、 Line 241: “Given that there are many normal modes of the Earth that have periods longer than 10 yr [40,41], signals with such periods could also be present in the GPS time series; consequently, the highest spectral peaks in Figures 7a and 7b and the first significant spectral peaks in Figure 7c might be caused by such overlapping long-period signals.” This bold statement is likely speculate. First, Earth’s global mode is usually absorbed by 7-parameter transformation during the reference frame alignment. Hence it is hard to be left in the site coordinate time series. Second, low frequency systematic errors cannot be excluded.
Response: Thanks for your suggestion. We do not sure where the long period terms come from, but referring that the 18.6yr tidal signal also considered in the tidal model when pre-process the GPS time series, we gave such suggestion in the last text. According to your suggestion, we modified the related expressions. Please see lines 246-248.
4、 Figure 7 and Figure 10, pattern inside the dash circle: Both spectrum show 2.5 years quasi-oscillation. But the wavelet analyses indicate negative correlation. How to explain this?
Response: Here we re-plot Figure 7f and Figure 10c together, please see the following figure.
We can see that the phases of the two spectra in the 2006-2011 timespan are consistent with each other quiet well. As the wavelet spectrum has end effects, so the results before ~2004 and after ~2014 in Figure 7f (the up subfigure here) will be affected by the drags of the long-period signals (such as the ~6yr signal). While for the spectrum in Figure 10c, the results also will be affected by the end effects (but the longer period signals in Figure 10c are quite weak, or said that no ~6yr signal in it). So if we only compare the results around the 2004 epoch, it seems that they do have negative correlation, but the useful information should be only referred from the 2006-2012 timespan.
Therefore, we further confirm that the ~2.5-3yr quasi-oscillations denoted by the ellipses in Figure 7f and Figure 10c have the positive correlation.

Reviewer 2 Report
Thank you for considering my comments and edits in revising your manuscript, it looks much better and is much clearer to me now. I only have a few additional (minor) comments, questions and edits. I believe your manuscript should be ready to publish after considering those.
Response to Comments:
Lines 52-53: Please clarify what you mean by non-tectonic vs crustal deformation here
Response: Done as you significant suggestion. Here non-tectonic deformation (e.g. surface hydrology, atmosphere and non-tidal ocean loading displacement). See in revised manuscript. Thanks.
->Non-tectonic deformation is also crustal deformation. Maybe use non-tectonic vs tectonic deformation instead.
Lines 80-81: Are there any data gaps in the time series? Are you using all stations available or only a subset?
Response: Thanks for your significant reminding. We have improved the data gaps by using linear interpolation
->How long are these data gaps? Linear interpolation might not be adequate for significantly long data gaps.
Lines 176-178: How do the seasonal (annual + semi-annual) signals that you remove here compare to those removed in the GRACE-derived datasets? Are they in phase and of similar amplitude? Could you add a sentence explaining why you didn’t keep the seasonal signals for the PCA.
Response: Before decomposing the CMC from regional GPS time series, the seasonal (annual + semi-annual) signals have been removed in single GPS time series. The seasonal oscillations are well known signals; however the signals in CMC are mainly related to long-periodic signals which are not unsolved of geophysical mechanism. We have improved this statement in revised version. Thanks.
->No need to discuss this in the paper but, as a side note, I would disagree that seasonal signals are well characterized signals. How to accurately extract them from GPS time series is still a topic of active research.
Detailed Comments:
Line 53: always
Line 55: Geodetic Measurements from remote sensing
Line 62: the water storage balance
Line 63: form generate
Line 66: Add a sentence to define what CMC is. The sentence on line 408-410 does a great job at that, why not place it earlier on?
Line 68: decompose a extract
Line 71: only
Line 77: GRACEdata constrained -derived
Line 94: The 3D GPS velocities were estimated in the ITRF 2008 reference frame using (…)
Line 114/115: invert compute, inversion computation
Line 117: replaced estimated
Line 127: at in the ETP
Line 154: spectral spectrum
Line 182-188: You could integrate these results in Figure 3 to make it easier to visualize, i.e., indicate the average spatial response of each PC in the relevant panel of Figure 3.
Line 193-194: I am not sure I understand this sentence, please rephrase.
Line 233:Wwhat do you mean by quasi-oscillation?
Line 233: (…) approximately 1 year. As we (…)
Line 240: What exactly do you mean by the signals overlap each other?
Line 284: Variations in surface loads due to Earth’s fluid envelope deform the elastic lithosphere. This deformation is captured by GPS, especially in the vertical component.
Line 297-301: This is true for GRACE-derived displacements using Swenson’s degree-1 term. However these papers show that the discrepancies between GPS and GRACE-derived horizontal displacements can be mostly reconciled by using a degree-1 contribution derived from the GPS dataset.
Line 302-303: Please specify exactly which signals you are referring to here.
Line 335-337: I am not sure I fully understand what you mean here.
Line 337-338: “Therefore, global climate change also has a significant effect on the interannual oscillations of CMCs.” This makes it sound like climate change and hydrological loading are two separate contributions to the CMCs, please rephrase.
Line 344: After correcting the GPS time series for the CMCs, we (…)
Line 347: (…) after filtering out the CMCs (…)
Line 352: (…) which reveal the effect of CMC on (…)
Line 376: motion deformation
Lines 416 – 419: This is not a complete sentence.
Line 452: predominantly most likely
Line 453: How do you know that they are nonstationary?
Line 456: with a period of 6 years
Line 460: displacements
Figures:
Figure 5c: There still appears to be a strong annual or quasi-annual signal in PC1. Yet there is no clear peak in the Fourier spectrum around the 1 year frequency. Could you explain why that is?
Figure 7 caption: Add a sentence to describe what the ellipse and the dashed vertical lines mean.
Figure 8 caption: Explain what the grey band is.
Figure 9j: Why is there still a peak at 1 year if you removed the annual signal?
Author Response
Response to Reviewer #2
Thank you for considering my comments and edits in revising your manuscript, it looks much better and is much clearer to me now. I only have a few additional (minor) comments, questions and edits. I believe your manuscript should be ready to publish after considering those.
Response to Comments:
Lines 52-53: Please clarify what you mean by non-tectonic vs crustal deformation here
Response: Done as you significant suggestion. Here non-tectonic deformation (e.g. surface hydrology, atmosphere and non-tidal ocean loading displacement). See in revised manuscript. Thanks.
->Non-tectonic deformation is also crustal deformation. Maybe use non-tectonic vs tectonic deformation instead.
Response: Done as your significant suggestion. Please see line 52.
Lines 80-81: Are there any data gaps in the time series? Are you using all stations available or only a subset?
Response: Thanks for your significant reminding. We have improved the data gaps by using linear interpolation
->How long are these data gaps? Linear interpolation might not be adequate for significantly long data gaps.
Response: Yes, we fully agree with you. In data pre-processing, we only chose the records with no gap longer than 7 days, such little gap has effect for further analysis, so we simply use a linear interpolation. We have added an explanation about this.
Lines 176-178: How do the seasonal (annual + semi-annual) signals that you remove here compare to those removed in the GRACE-derived datasets? Are they in phase and of similar amplitude? Could you add a sentence explaining why you didn’t keep the seasonal signals for the PCA.
Response: Before decomposing the CMC from regional GPS time series, the seasonal (annual + semi-annual) signals have been removed in single GPS time series. The seasonal oscillations are well known signals; however the signals in CMC are mainly related to long-periodic signals which are not unsolved of geophysical mechanism. We have improved this statement in revised version. Thanks.
->No need to discuss this in the paper but, as a side note, I would disagree that seasonal signals are well characterized signals. How to accurately extract them from GPS time series is still a topic of active research.
Response: Thanks for your significant suggestion. We have removed this part in revised manuscript. We are agreeing with you. The seasonal signals are still a topic of active research in GPS time series but they are not components in CMCs.
Detailed Comments:
Line 53: always
Response: Done as you significant suggested.
Line 55: Geodetic Measurements from remote sensing
Response: Done as you significant suggested.
Line 62: the water storage balance
Response: Done as you significant suggested.
Line 63: form generate
Response: Done as you significant suggested.
Line 66: Add a sentence to define what CMC is. The sentence on line 408-410 does a great job at that, why not place it earlier on?
Response: CMC is means common mode component, we have removed relate sentence to line 66-69. Thanks for your significant suggestion.
Line 68: decompose a extract
Response: Done as you significant suggested.
Line 71: only
Response: Done as you significant suggested.
Line 77: GRACEdata constrained -derived
Response: Done as you significant suggested.
Line 94: The 3D GPS velocities were estimated in the ITRF 2008 reference frame using (…)
Response: Done as you significant suggested.
Line 114/115: invert compute, inversion computation
Response: Done as you significant suggested.
Line 117: replaced estimated
Response: Done as you significant suggested.
Line 127: at in the ETP
Response: Done as you significant suggested.
Line 154: spectral spectrum
Response: Done as you significant suggested.
Line 182-188: You could integrate these results in Figure 3 to make it easier to visualize, i.e., indicate the average spatial response of each PC in the relevant panel of Figure 3.
Response: Done as you significant suggested. Please see in Figure 3 in revised manuscript.
Line 193-194: I am not sure I understand this sentence, please rephrase.
Response: Done as you suggested. It has been rephrase as “Eigenvectors are arranged according to the data size, which consist with the powers of eigenvalues from low to high orders.”
Line 233:What do you mean by quasi-oscillation?
Response: The quasi-oscillation means the “Non-quasi-periodic”. We delete “quasi-” in order to avoid statement confusion.
Line 233: (…) approximately 1 year. As we (…)
Response: Done as you significant suggested.
Line 240: What exactly do you mean by the signals overlap each other?
Response: It means that there are some signals with periods longer than 6yr, but the length of the used records is limited, so their spectral peaks will overlap with each other in the frequency domain/or in the wavelet spectrum. We changed the expressions, please see lines 249-251.
Line 284: Variations in surface loads due to Earth’s fluid envelope deform the elastic lithosphere. This deformation is captured by GPS, especially in the vertical component.
Response: Done as you significant suggested. Please see lines 300-301.
Line 297-301: This is true for GRACE-derived displacements using Swenson’s degree-1 term. However these papers show that the discrepancies between GPS and GRACE-derived horizontal displacements can be mostly reconciled by using a degree-1 contribution derived from the GPS dataset.
Response: Thanks for your useful comments. Our results just show a local result, there are still lots of works need to do for the relationships between on GPS and GRACE datasets. According to your comments, we modified the related expressions, please see lines 335-339.
Line 302-303: Please specify exactly which signals you are referring to here.
Response: Done as you suggested. It refers to Figure 9k and 9l.
Line 335-337: I am not sure I fully understand what you mean here.
Response: It has been rephrased as “Simultaneously, the CMCs and hydrologic loading deformation show anomalous oscillations during this period, which are response to global climate change.”
Line 337-338: “Therefore, global climate change also has a significant effect on the interannual oscillations of CMCs.” This makes it sound like climate change and hydrological loading are two separate contributions to the CMCs, please rephrase.
Response: Done as you suggested. It has been rephrased as “Therefore, global climate change also has a potential effect on regional hydrological loading that is manifested by interannual oscillation of CMCs.” See in revised manuscript. Please see lines 353-356.
Line 344: After correcting the GPS time series for the CMCs, we (…)
Response: Done as you significant suggested.
Line 347: (…) after filtering out the CMCs (…)
Response: Done as you significant suggested.
Line 352: (…) which reveal the effect of CMC on (…)
Response: Done as you significant suggested.
Line 376: motion deformation
Response: Done as you significant suggested.
Lines 416 – 419: This is not a complete sentence.
Response: It has been improved in revised manuscript. Thanks.
Line 452: predominantly most likely
Response: Done as you significant suggested.
Line 453: How do you know that they are nonstationary?
Response: We delete “nonstationary” in order to avoid confusion statement.
Line 456: with a period of 6 years
Response: Done as you significant suggested.
Line 460: displacements
Response: Done as you significant suggested.
Figures:
Figure 5c: There still appears to be a strong annual or quasi-annual signal in PC1. Yet there is no clear peak in the Fourier spectrum around the 1 year frequency. Could you explain why that is?
Response: Thanks for you careful review. We rechecked the results, and find that the look-like annual signal just the envelope of some high frequency noise. The following Figure R1a shows the PC1 and its filtered time series, here we use a low-pass filter (signal with f>1.5cpy with be filtered). R1c shows the Fourier spectra of these two time series in R1a, we can see that the spectral peaks for the annual signal are not change before and after the filter process, while the look-like ‘annual signal’ is not presented in the filtered time series any more. Figure R1c shows a segment of the Figure R1a to more clearly show that the look-like ‘annual signal’ just some high frequency noise. As the reason of such high frequency noise, we have no more explanation, they may be caused by the station itself.
According to your comments, we add some explanations, please see Lines 219-225.
Figure R1. (a) The PC1 shown in Figure 5c and its filtered time series; (b) the Fourier spectra of the two time series in (a); (c) a segment of (a).
Figure 7 caption: Add a sentence to describe what the ellipse and the dashed vertical lines mean.
Response: Done. Please see lines 290-291.
Figure 8 caption: Explain what the grey band is.
Response: Done. Please see lines 294-298.
Figure 9j: Why is there still a peak at 1 year if you removed the annual signal?
Response: Thanks for your careful review. The results shown in Figure 9i, 9j, 9k and 9l all show that there are still have some energy around the 1 yr period. Such results caused by two reasons: the first one is that the east components have different noise with the north components (the same scale will affect the display of results); the second one is that there are still some other signals with periods close to 1yr contained in the east components. Here we enlarge the frequency axis of Figure 9j, see the following Figure R2. We can see that there is a significant spectral peak corresponds to ~1.26cpy frequency, and the amplitude of it is ~0.047mm; there also has a spectral peak corresponds to ~0.97cpy frequency with a ~0.034mm amplitude. From Figure 9a, we can see that the amplitudes of the annual signals are ~0.2-0.4mm, which are about 10 times than the residual amplitudes in Figure 9j. As we removed the annual signal with a 1.0 cpy frequency, the ~1.26cpy signal will not be affected and still presented in the residual time series. Although the spectral peak with ~0.97cpy frequency in Figure 9j looks quite significant, comparing with the longer frequency band as shown in Figure R2, it is just close to the background noise and cannot be identified as a significant peak.
In short, the annual signals with ~0.2-0.4mm amplitude have almost been completely fitted and removed (of course there has some fitting deviations), the residual signals around the ~1 yr period are close to the background noise level, and not significant any more.
Figure R2, the Fourier spectra same as Figure 9j, but with a longer frequency band.
